# Stress Factors as Possible Regulators of Pluripotent Stem Cell Survival and Differentiation

**DOI:** 10.3390/biology12081119

**Published:** 2023-08-11

**Authors:** Toqa Darwish, Nuha Taysir Swaidan, Mohamed M. Emara

**Affiliations:** Basic Medical Sciences Department, College of Medicine, QU Health, Qatar University, 2713 Doha, Qatar

**Keywords:** embryonic stem cells, induced pluripotent stem cells, stress response, cell survival, differentiation

## Abstract

**Simple Summary:**

The unique capacity of pluripotent stem cells to differentiate into various types of cells elucidates their significant importance. Many research studies focused on understanding the pluripotency and differentiation aspects of those cells. However, the fundamental question of how pluripotent stem cells manage to survive stressful conditions and the effects of various environmental stressors on them is yet to be answered. Therefore, in this review, we aim to cover the critical role of stress response elements and their possible role in regulating stem cell growth and differentiation.

**Abstract:**

In recent years, extensive research efforts have been directed toward pluripotent stem cells, primarily due to their remarkable capacity for pluripotency. This unique attribute empowers these cells to undergo self-renewal and differentiate into various cell types originating from the ectoderm, mesoderm, and endoderm germ layers. The delicate balance and precise regulation of self-renewal and differentiation are essential for the survival and functionality of these cells. Notably, exposure to specific environmental stressors can activate numerous transcription factors, initiating a diverse array of stress response pathways. These pathways play pivotal roles in regulating gene expression and protein synthesis, ultimately aiming to preserve cell survival and maintain cellular functions. Reactive oxygen species, heat shock, hypoxia, osmotic stress, DNA damage, endoplasmic reticulum stress, and mechanical stress are among the examples of such stressors. In this review, we comprehensively discuss the impact of environmental stressors on the growth of embryonic cells. Furthermore, we provide a summary of the distinct stress response pathways triggered when pluripotent stem cells are exposed to different environmental stressors. Additionally, we highlight recent discoveries regarding the role of such stressors in the generation, differentiation, and self-renewal of induced pluripotent stem cells.

## 1. Introduction

Pluripotent stem cells (PSCs) possess a remarkable ability to self-renew and differentiate into various cell types originating from the three germ layers; hence, their name “pluripotent” is derived from the Latin roots of pluri (meaning many) and potent (meaning power) [1]. Two examples of PSCs are embryonic stem cells (ESCs) and induced pluripotent stem cells (iPSCs). ESCs are isolated from the inner cell mass of a blastocyst during the preimplantation stage of a fertilized ovum [1]. On the other hand, iPSCs are ESC-like cells generated through the reprogramming of somatic cells in a laboratory setting [2]. Maintaining a delicate balance is imperative for regulating the self-renewal and differentiation processes in these cells, as it directly impacts their survival and functional integrity. Several crucial factors have been identified to play a significant role in the regulation of PSCs. Analysis of the molecular profile of PSCs has revealed distinct gene regulatory processes involving a diverse range of regulatory protein factors and their target genes [3]. These protein factors encompass a range of molecules, including signaling pathways components and transcription factors, which play a pivotal role in coordinating the processes of self-renewal and differentiation. Additionally, the PSCs microenvironment is recognized as a major determinant of PSCs’ survival, stability, and activity [4]. Any event occurring within this niche that hampers the accumulation of stem cells or impairs their ability to differentiate and produce normal and sufficient parenchymal products is defined as stress [5].

PSCs can be affected by two distinct types of stresses: extrinsic and intrinsic stresses. Extrinsic stress refers to any detrimental change in the biological system caused by adverse environmental factors. On the other hand, intrinsic stress arises from metabolic challenges such as the accumulation of waste products, generation of reactive metabolites, or DNA alterations resulting from repeated cell division [5]. These stressors interact with various cellular components, triggering both acute and chronic adaptation mechanisms collectively known as the cellular stress response. Such a response is essential for cell survival, and any disruption in its functioning can contribute to disease development or cellular apoptosis. Consequently, PSCs employ different stress response mechanisms to ensure their survival under severe stressful conditions [4]. The exceptional capability of PSCs to differentiate into diverse cell types has positioned them as an irreplaceable tool in both disease modeling and potential medical therapies. Human-derived PSCs, in particular, have offered invaluable insights into the pathophysiology of specific diseases by serving as reliable disease models. Moreover, the utilization of patient-specific iPSCs has provided a safer approach for screening potential medical therapies and assessing their toxicological effects [6]. Given the paramount importance of PSCs, numerous research studies have been dedicated to unraveling the mechanisms underlying their pluripotency and differentiation. However, a fundamental question that remains incompletely understood is: how do PSCs navigate through stressful conditions and maintain their survival? Hence, the aim of this review is to delve into the crucial role of stress response elements and their potential influence on governing stem cell survival and determining cell fate. By shedding light on this aspect, we strive to deepen our comprehension of the intricate mechanisms that enable PSCs to withstand and thrive under challenging circumstances.

## 2. The Role of Stress Factors in Embryonic Growth and Proliferation

When confronted with stressors, PSCs activate a variety of stress response mechanisms to maintain their survival under such challenging conditions. These mechanisms include various protein kinases that mediate the stress response, which exhibit functions in both the cytosol and nucleus. In the face of elevated stress levels, stress enzymes modulate the activity of diverse transcription factors, which play a significant role in orchestrating the stress response by regulating the expression of a broad array of genes [7]. In vivo, different stressors have been identified to adversely impact the embryonic development. Hyperosmotic stress resulting from dehydration or diabetes, hypoxia, heat shock, and exposure to environmental toxins such as metals, dioxins, or benzopyrene are among the most commonly known stressors. Conversely, in vitro settings allow for the induction of different stressors such as shear stress during handling as well as those resulting from cryopreservation, imbalanced oxygen levels, or culture media composition and conditions [8]. Notably, extensive research has focused on hypoxic, osmotic, mechanical and oxidative stress as the most studied stressors. In this discussion, we explore these stressors, elucidate their associated cellular responses, and evaluate their significance in the context of ESC development (Table 1).

### 2.1. Hypoxic Stress

Hypoxic stress refers to the physiological condition where cells or tissues experience reduced oxygen availability or oxygen levels below the norm, leading to cellular responses and adaptations. This stress can be induced in experimental settings by deliberately lowering oxygen levels below atmospheric conditions (typically around 20% oxygen) to create an environment with an oxygen concentration of approximately 1–5%. Additionally, hypoxic stress is often studied in the context of cellular physiology, disease pathogenesis, and therapeutic interventions [15].

Indeed, hypoxia has been associated with the enhanced proliferation rate of trophoblastic stem cells (TSCs). Culturing mouse TSCs (mTSCs) under hypoxic conditions at 2% oxygen concentration resulted in a significantly higher growth rate compared to cultures maintained at 0%, 0.5%, or 20% oxygen [9]. On the molecular level, hypoxia activates the stress-activated protein kinase (SAPK), which is a known enzyme involved in stress response pathways. Once activated, SAPK is phosphorylated and phosphorylates various downstream targets, including transcription factors and other signaling molecules. This phosphorylation cascade leads to the activation of specific cellular responses, such as cell survival, proliferation, apoptosis, and adaptation to hypoxic conditions [9]. Interestingly, the levels of phosphorylated SAPK were found to be the lowest in mTSCs cultured at 2% oxygen, demonstrating a potential attenuation of cellular stress [9]. Supporting this observation, levels of the cleaved caspase 3 protein, which is indicative of apoptotic activity, were highest in mTSCs cultured at 0% oxygen and have decreased progressively with increasing oxygen levels, particularly at 2% oxygen cultures. Examining the expression of nuclear multipotency factors, namely CDX2, Id2, and estrogen-related receptor β (Errβ), revealed their highest expression in mTSCs cultured at 2% and 20% oxygen, with a significant reduction observed at 0.5% oxygen cultures. Conversely, the expression levels of differentiation markers, including Glial cells missing (Gcm1), trophoblast-specific protein α (Tpbpa), and heart and neural crest derivatives expressed 1 (Hand1), were significantly lower in mTSCs cultured at 2% and 20% oxygen compared to 0.5% oxygen cultures. These findings suggest that culturing mTSCs at 2% oxygen provides a favorable environment for maintaining multipotency, while 0.5% oxygen promotes differentiation [9].

Similar results were observed in human TSCs (hTSCs) cultured at 2% or 20% oxygen, where cell counts were higher in hTSCs cultured at 2% oxygen, indicating a stimulatory effect of hypoxia on cellular proliferation. Further analysis revealed that hTSCs cultured at 2% oxygen expressed higher levels of cyclin B (a mitosis promoting protein which serves as a proliferation marker) compared to hTSCs cultured at 20% oxygen. In contrast, the expression of the cell cycle regulator, cyclin-dependent kinase inhibitor 1 (p21Cip1/Waf1), was significantly higher in hTSCs cultured at 20% oxygen, indicating cell cycle arrest. These findings suggest that hTSCs cultured under hypoxic conditions are induced to enter mitosis, while cells at higher oxygen levels experience cell cycle arrest [10].

### 2.2. Osmotic Stress

Osmotic stress refers to the physiological condition where cells are exposed to changes in osmolarity, resulting in alterations in cell volume and water balance. It can occur due to imbalances in solute concentration across cell membranes or exposure to external osmotic agents. Osmotic stress can have significant impacts on cellular processes and function [11]. In contrast to hypoxia, hyperosmotic stress has shown to adversely affect the growth rate of cultured human and mouse TSCs. Treatment of cells with 50 mM sorbitol was able to significantly decrease the number of cultured cells, while both 100 mM and 200 mM sorbitol treatment resulted in a highly significant reduction in cell number compared to untreated cells. However, the exposure of cells to 600 mM sorbitol led to a complete loss of cell viability [11]. The levels of phosphorylated SAPK in cells exhibited a dose-dependent increase when cells were exposed to sorbitol concentrations ranging from 50 to 600 mM for 30 min [11]. Similar findings were noticed in mouse TSCs, where the incidence of apoptosis increased with higher sorbitol concentrations [16]. Such results suggest that hyperosmotic stress by sorbitol induces apoptosis events and elevates pSAPK levels in TSCs.

### 2.3. Mechanical Stress

Mechanical stress in cell culture refers to the physical forces or deformations that cells experience due to external mechanical stimuli [17]. Shear force is one of these factors that stimulate mechanical stress and refers to the mechanical forces exerted on cells by fluid flow or mechanical agitation, which can impact various cellular processes including gene expression, cell adhesion, migration, and mechanotransduction pathways [12]. Notably, shear stress has been found to reduce the number of cells in cultured embryos. The exposure of mouse embryos to shear stress of 1.2 dynes/cm_2_ for 12 h was sufficient to cause lethality to these blastocysts and significantly increase the phosphorylation levels of mitogen-activated protein kinase 8/9 (MAPK8/9, which is known as a stress-activated protein kinase/junC kinase 1/2). When embryos cultures were treated with MAPK8/9 phosphorylation inhibitor, 50% of apoptosis events were restrained, proposing a causal role for MAPK8/9 phosphorylation in the shear stress-induced lethality. Furthermore, lower shear stresses could stimulate adequate MAPK8/9 phosphorylation that would decelerate growth [12].

### 2.4. Oxidative Stress

Chemical stress refers to the exposure of cells to chemical substances that can disrupt normal cellular functions, leading to physiological or pathological changes. It involves the impact of chemicals, such as pollutants, toxins, and drugs, that can lead to cellular toxicity, oxidative stress, alterations in gene expression and cellular signaling pathways, which ultimately impact cellular homeostasis and viability [4]. The oxidative stress is related to an imbalance between the production of reactive oxygen species (ROS) and the ability of the cells to neutralize them, resulting in cellular damage and potential harm to various biological processes. ROS are natural by-products of cellular metabolism; however, the overproduction of ROS is associated with cell exposure to chemical stressors [4].

Chemical stress has induced apoptosis in embryonic stem cells (ESCs), as observed in zearalenone-exposed embryonic bodies. Zearalenone, a chemical toxin, induced toxicity through ROS production. Increasing zearalenone concentrations led to a dose-dependent increase in cell arrest at the G0/G1 phase and apoptosis rate. In addition, this chemical stress inhibited cell proliferation, elevated ROS levels, disrupted mitochondrial membrane potential, and upregulated apoptosis markers, including p53, Bax, caspase-9, and caspase-3. These findings were confirmed by the pretreatment of ESCs with the antioxidant Trolox that attenuated the zearalenone-induced ROS generation, apoptosis rate, and the expression of p53, Bax, and caspase-9 [13].

In a study investigating the impact of ROS exposure on human embryonal carcinoma NT2 cells, two different agents were utilized: paraquat and hydrogen peroxide (H_2_O_2_). When NT2 cells were exposed to paraquat at increasing concentrations, pluripotency markers such as Nanog, Oct4, and teratocarcinoma-derived growth factor 1 (Tdgf1) showed a clear dose-dependent downregulation. However, intriguingly, the introduction of H_2_O_2_ had a much milder effect on pluripotency markers expression, with only a slight reduction observed in Oct4 expression at higher concentrations, while the overall pluripotency marker levels remained largely unchanged. Additionally, the researchers performed experiments with different durations of paraquat treatment on NT2 cell cultures. Interestingly, they found that as the treatment time increased, there was a corresponding increase in the expression of neuronal differentiation markers, specifically, Pax6, GDNF family receptor alpha 1 (GFRα1), homeobox A1 (Hoxa1), and neural cell adhesion molecule 1 (Ncam1). In contrast, the expression of cytochrome P450 family 26 subfamily A1 (Cyp26a1) decreased with longer treatment times, and this is particularly noteworthy since Cyp26a1 negatively regulates the all-trans retinoic acid (atRA) signaling pathway responsible for neuronal differentiation [14,18]. Overall, these findings provide valuable insights into the potential role of ROS in influencing the fate of hESCs, shedding light on the intricacies of pluripotency regulation and neuronal differentiation pathways.

## 3. Potential Stress Responses Mechanisms in PSCs

The embryonic stress response is a crucial mechanism that enables cells to survive and undergo differentiation in the presence of high stress levels [7]. Although the underlying mechanisms are not fully elucidated, numerous studies have explored stress response pathways in various eukaryotic cells. These pathways have been confirmed in different cell types, including hepatoma cells, embryonic fibroblasts, human embryonic kidney 293 cells, peritoneal macrophages, HeLa cells, K-562 cells, cardiovascular cells, DU-145 cells, osteoblasts, U-2 OS human osteosarcoma cells, PC12 cells, and CHO-K1 cells, suggesting that PSCs may exhibit similar stress response pathways when subjected to stress [7]. The cellular stress response involves a group of stress enzymes from the protein kinase family, including SAPKs, MAPKs (e.g., p38), AMP-activated protein kinase (AMPK), phosphoinositide 3-kinases (PI3K), protein kinase B (PKB or Akt), mitogen-activated protein kinase kinases (MEK1/2, MEKK4), protein kinase A (PKA), inositol-requiring enzyme 1 (IRE1), and protein kinase R (PKR)-like endoplasmic reticulum kinase (PERK). These enzymes primarily function in the cytosol under low-stress conditions but translocate to the nucleus under higher stress levels to modulate the activity of transcription factors [8]. Transcription factors involved in the stress response include MAPK-activated protein kinases (MAPKAPs), activating transcription factor 4 (ATF4), X-box binding protein 1 (XBP1), Oct4, hypoxia-inducible factors (HIFs), nuclear factor erythroid-derived factor 2 (Nrf2/KEAP2), nuclear factor kappa-light-chain-enhancer of activated B cell (NFκB), nuclear factor of activated T-cells (NFAT5), heat shock factor 1/2 (HSF1/2), and potency factors such as inhibitor of binding/differentiation (Id2), caudal-type homeodomain protein 2 transcription factor (Cdx2), Sox2, Nanog, and Rex1. These transcription factors regulate the expression of various genes implicated in the cell survival process, including those which lead to the prioritized differentiation of ESCs into vital cell lineages before less essential lineages [8]. Many of these transcription factors are linked to a specific type of stress. For instance, NRF2 is involved in oxidative stress, HSF1 is associated with heat shock response, p53/TRP53 responds to DNA damage, HIF1 is activated under hypoxic conditions, XBP1, ATF4, and ATF6 are implicated in endoplasmic reticulum (ER) stress, metal regulatory transcription factor 1 (MTF1) is responsive to metal stress, NFκB is involved in inflammatory stress, and NFAT5 is associated with hypertonic stress [4]. Given the potential similarities in stress response mechanisms in both eukaryotic cells and PSCs, we will delve into the various stress responses observed in eukaryotic cells, with a particular focus on the cellular responses to oxidative stress, heat stress, hypoxic stress, osmotic stress, p53-mediated stress, and endoplasmic reticulum stress (Figure 1). We illustrate these different mechanisms of stress responses in Figure 2, Figure 3, Figure 4, Figure 5, Figure 6 and Figure 7.

### 3.1. Oxidative Stress Response

Under normal cellular conditions, Nrf2 is complexed with Kelch-like ECH-associated protein 1 (Keap1) (that is rich in cystine residues) forming the Keap1–Nrf2 complex, which is sequestered in the cytoplasm by its interaction with the actin cytoskeleton. However, when cells are exposed to oxidative stress, several sensors such as MAPKs, extracellular signal-regulated kinases (ERK), p38, protein kinase C (PKC), and PI3K are activated, leading to the phosphorylation of both Keap1 and Nrf2 [19,20]. This phosphorylation event along with the direct interaction of cystine residues of Keap1 with ROS induces conformational changes in Keap1, reducing its affinity to Nrf2, which becomes stable and active when phosphorylated [21]. Consequently, Nrf2 becomes liberated and able to translocate into the nucleus where it forms heterodimers with musculoaponeurotic fibrosarcoma oncogene (Maf) proteins. The resulted Nrf/maf complex displaces basic leucine zipper transcription factor 1 (Bach1) (a transcriptional repressor), and it interacts directly with the antioxidant response elements (AREs) [22]. This leads to the upregulation of oxidative stress response genes including heme oxygenase-1 (HMOX1), glutathione S-transferase A2 (GSTA2), and NADPH quinone oxidoreductase (NQO1). These genes play essential roles in ROS elimination, glutathione synthesis, and the removal of oxidized proteins, thereby protecting the cell from oxidative damage [21,22,23]. NRF2 deactivation occurs through the formation of the Keap1–Roc1–Cul3 complex, which recruits E2 ubiquitin ligase to degrade Nrf2 [24]. Normally, Nrf2 has a relatively short half-life of approximately 20 min under non-stress conditions [25].

**Figure 2 biology-12-01119-f002:**
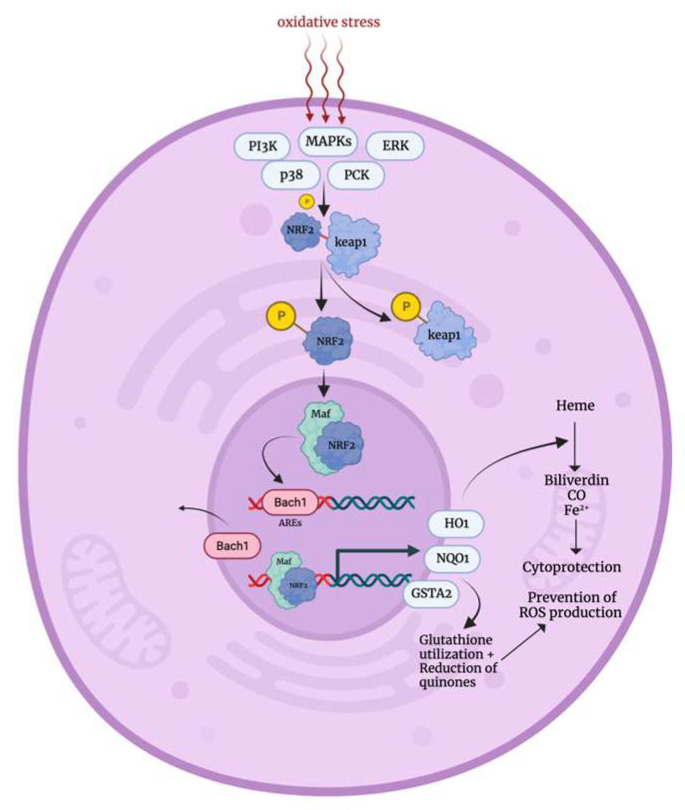
Oxidative stress mechanism [19,20,21,22]. PIK3: phosphatidylinositol 3 kinase, MAPKs: mitogen activated protein kinases, ERKs: extracellular signal-regulated kinases, PCK: protein kinase C, NRF2: nuclear factor elytroid-derived factor 2, Keap1: kelch ECH-associating protein 1, Maf: Musculoaponeurotic Fibrosarcoma Oncogene proteins, Bach1: Basic Leucine Zipper Transcription Factor 1, AREs: antioxidant response elements, HO1: heme oxygenase-1, NQO1: NADPH quinone oxidoreductase, GSTA2: Glutathione S-transferase A2, CO: carbon monoxide.

### 3.2. Heat Shock Stress Response

The heat shock response pathway is a crucial mechanism activated in response to various physical and chemical stressors, including elevated temperatures, chemical toxins, heavy metals, oxidative stress, as well as conditions such as infections, inflammations, cancers, ischemia and neurodegenerative diseases. This activation upregulates the heat shock proteins expression [26]. Normally, heat shock transcription factor 1 (Hsf-1) is maintained in its inactive state in the cytoplasm through its association with heat shock protein 90 (Hsp90) [27]. Nevertheless, under stress conditions, Hsf-1 rapidly undergoes trimerization and translocates to the nucleus, where it becomes activated through phosphorylation by calcium/calmodulin-dependent protein kinase II (CaMKII) and protein kinase casein kinase II (CK2) [28,29,30]. Subsequently, the activated Hsf-1 binds to heat shock response elements in promoter regions, leading to the upregulation of target genes [31]. These target genes include chaperone proteins (e.g., Hsp70, Hsp27), interleukin 6, and multidrug resistance protein 1, which are involved in eliminating damaged tissues, foreign material, and toxic metabolites from the cell [23,30,32]. Lastly, Hsf-1 is deactivated by binding to heat shock binding protein 1 (Hsbp1) [33].

**Figure 3 biology-12-01119-f003:**
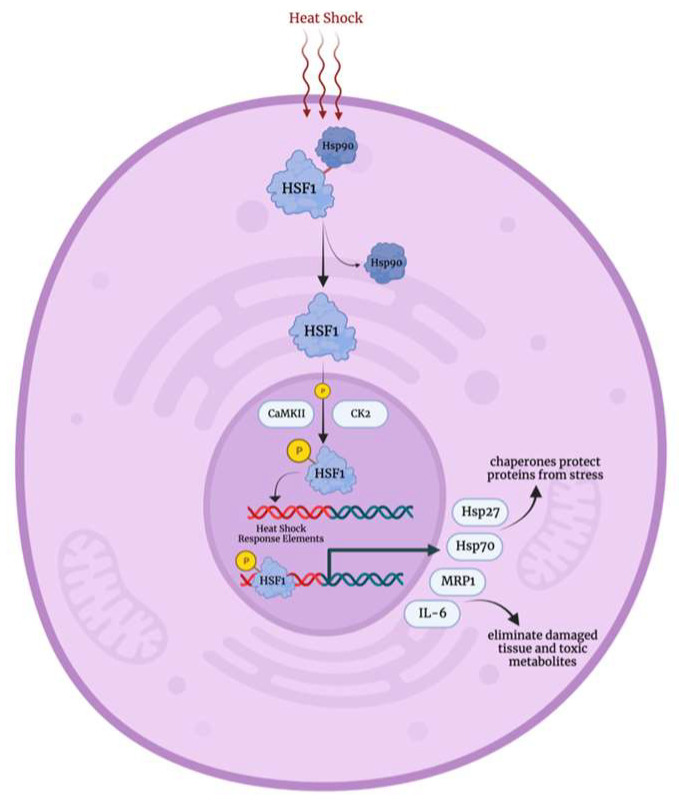
Heat shock stress response mechanism [27,28,29,30]. Hsp90: heat shock protein 90, HSF1: heat shock transcription factor 1, CaMKII: calcium/calmodulin-dependent protein kinase II, CK2: casein kinase II, Hsp27: heat shock protein 27, Hsp70: heat shock protein 70, MRP1: multidrug resistance protein 1, IL-6: interleukin 6.

### 3.3. Hypoxic Stress Response

The family of enzymes known as prolyl hydroxylases (PHD) constantly monitors the intracellular oxygen levels [34]. Under normal conditions, PHD proteins hydroxylate proline residues in HIF-1α, facilitating its direct interaction with von Hippel–Lindua tumor suppressor protein (VHL), which in fact leads to HIF-1α degradation [35]. On the other hand, in hypoxic conditions, impaired PHD activity and the involvement of signal transducers such as p38 and PI3K kinases lead to HIF-1α stabilization [36,37,38]. This allows HIF-1α to form a heterodimer with HIF-1β, which binds to hypoxia response elements (HREs) within gene promotor regions, promoting the transcription of target genes as well as recruiting transcriptional activators p300 and cAMP response element binding protein (CBP) [39,40]. These target genes are involved in oxygen transport, iron transport, angiogenesis, glycolysis regulation, and glucose uptake [41].

**Figure 4 biology-12-01119-f004:**
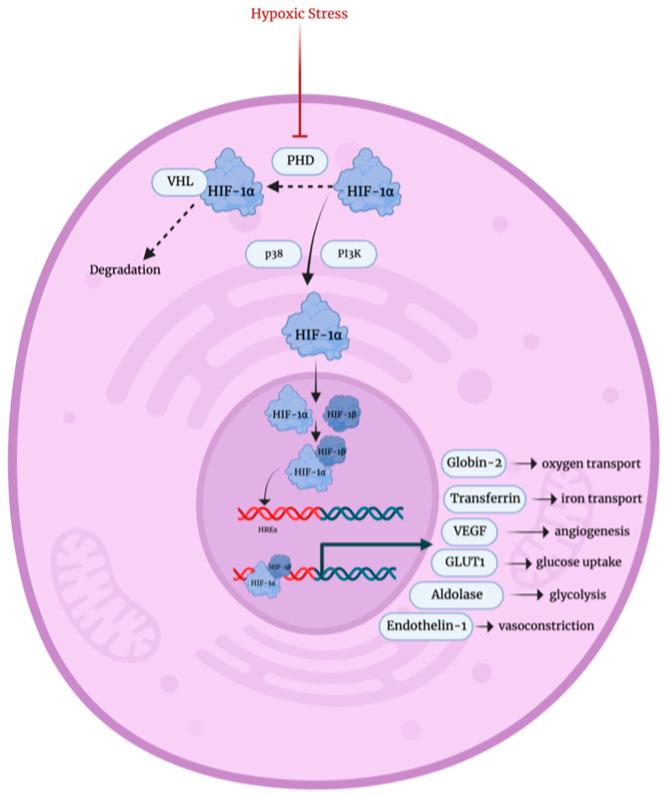
Hypoxic stress response mechanism [35,36,37,38,39,40,41]. PHD: prolyl hydroxylases, HIF-1α: hypoxia-inducible factor alpha subunit, VHL: von Hippel–Lindua tumor suppressor protein, PI3K: phosphatidylinositol 3 kinase, HIF-1β: hypoxia-inducible factor beta subunit, HREs: hypoxia response elements, VEGF: vascular endothelial growth factor, GLUT1: glucose transporter 1.

### 3.4. Osmotic Stress Response

The osmotic stress response pathway, primarily observed in renal medulla cells, remains one of the least understood stress response mechanisms. At the core of this pathway is the tonicity element binding protein (TonEBP/NFAT5), whose precise activation mechanism is not fully elucidated. In response to osmotic stress, TonEBP is believed to be activated by p38 kinase, Ras-related C3 botulinum toxin substrate 1 (Rac1), and possibly by ataxia-telangiectasia mutated (ATM) kinase and protein kinase A [42,43,44,45,46]. Upon activation, TonEBP translocates to the nucleus where it interacts with activator protein 1 and upregulates the expression of genes encoding solute transporters (urea transporter A1), organic osmolyte transporters (betaine GABA transporter and aldolase A), and chaperone proteins like Hsp70 [47,48].

**Figure 5 biology-12-01119-f005:**
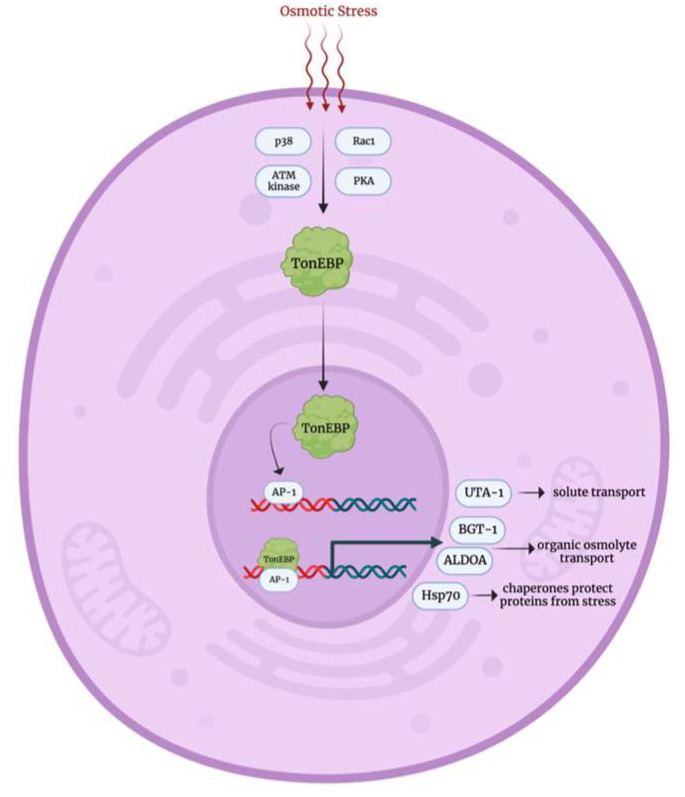
Osmotic stress response mechanism [42,43,44,45,46]. Rac1: Ras-related C3 botulinum toxin substrate 1, ATM kinase: ataxia-telangiectasia mutated kinase, PKA: protein kinase A, TonEBP: tonicity element binding protein, AP-1: activating protein-1, UTA-1: urea transporter A1, BGT-1: betaine GABA transporter, ALDOA: aldolase A, Hsp70: heat shock protein 70.

### 3.5. DNA Damage p53 Mediated Stress Response

In the absence of cellular stress, the tumor suppressor protein p53 is primarily located in the cytoplasm, and it is tightly bound to its negative regulator, murine double minute 2 (Mdm2), which leads to rapid degradation through ubiquitin-mediated pathways. This characteristic keeps the half-life of p53 relatively short [49]. However, when DNA damage occurs due to different factors such as chemical exposure, radiation, or hypoxia, a series of events is initiated to stabilize p53 and prevent its degradation. Phosphorylation of the transactivation domain of p53 by ATM, jun N-terminal kinase (JNK), and checkpoint kinases 1 and 2 (Chk1, Chk2) impedes the interaction between p53 and Mdm2 [49,50,51], while MAPKs and CK2 modify the carboxyl terminal of p53, enabling its tetramerization [52,53]. Once activated, p53 governs the expression of a wide array of genes. For instance, cyclin-dependent kinase inhibitor 1A (CDKN1A) and cyclin G orchestrate cell cycle arrest at the G1/S or G2/M phase [54]. Once growth is arrested, cells can proceed with DNA repair through the facilitation of DNA damage-inducible alpha (GADD45A) and xeroderma pigmentosum (XPC) or undergo apoptosis by the modulation of Bcl-2-associated X protein (Bax) and B-cell lymphoma-extra-large (Bcl-L) [55,56,57]. Interestingly, p53 also enhances the expression of its negative regulator, Mdm2, establishing a negative feedback loop [58].

**Figure 6 biology-12-01119-f006:**
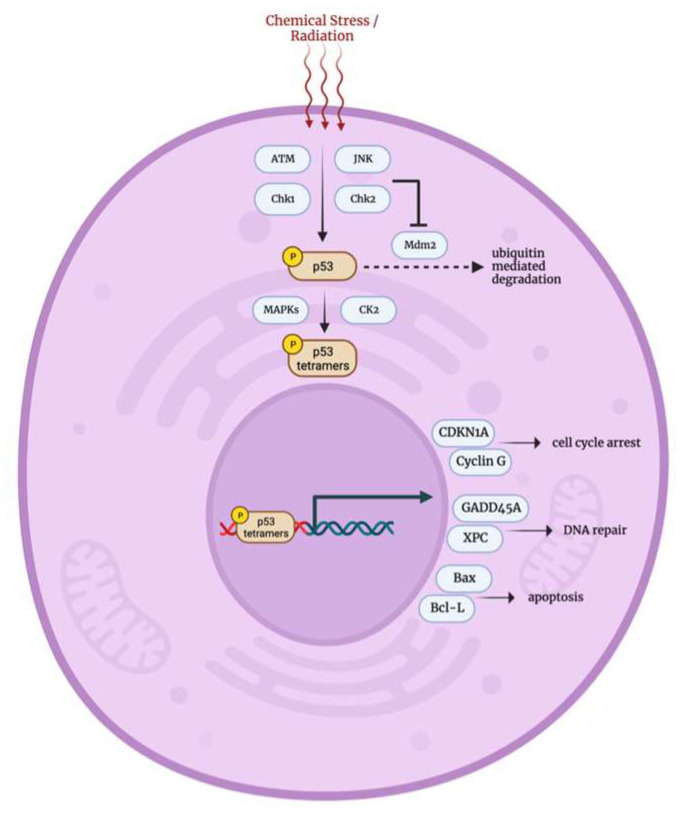
DNA damage p53 mediated stress response mechanism [55,56,57]. ATM: ataxia-telangiectasia mutated, JNK: jun N-terminal kinase, Chk1 and Chk2: checkpoint kinases 1 and 2, Mdm2: murine double minute 2, MAPKs: mitogen-activated protein kinases, CKs: casein kinases, CDKN1A: cyclin dependent kinase inhibitor 1A, GADD45A: growth arrest and DNA damage inducible alpha, XPC: xeroderma pigmentosum, Bax: Bcl-2-associated X protein, Bcl-L: B-cell lymphoma-extra large.

### 3.6. Endoplasmic Reticulum Stress Response

The endoplasmic reticulum (ER) is an essential organelle responsible for protein synthesis and modification [59]. ER stress can be triggered by chemical toxins such as tunicamycin and thapsigarin, which inhibits protein glycosylation and disrupts calcium-ATPase activity, respectively, leading to the accumulation of unfolded proteins and calcium depletion in the ER [59,60,61,62]. Under normal cellular conditions, the ER chaperone protein, binding immunoglobulin protein (Bip/Grp78), resides in the ER lumen and interacts with the ER kinase PERK, ER-specific protein IRE1α and ATF6. However, when cells are exposed to stress, three distinct stress response pathways are initiated.

The first pathway, Bip, dissociates from PERK in favor of binding accumulating unfolded proteins, causing PERK to autophosphorylate and form homodimers. Activated PERK then phosphorylates eukaryotic translation initiation factor 2 (eIF2) that is responsible for recruiting tRNA to the ribosome for protein translation. This phosphorylation disables eIF2 from binding to guanosine triphosphate (GTP) to initiate translation, leading to transient protein synthesis inhibition [60,63]. The second pathway, the dissociation of IRE1α from Bip, triggers the autophosphorylation and activation of IRE1α. Upon activation, IRE1α splices XBP1 mRNA, leading to the translation of XBP1 protein in the cytosol [64]. It is noteworthy that XBP1 mRNA is among the transcripts that undergo cytosolic translation independent of the termination of protein synthesis mediated by ER stress. Therefore, XBP1 protein translocates to the nucleus and upregulates the expression of ER stress-related genes including those encoding chaperone proteins such as p58 and ERdj4, which are implicated in the degradation machinery of multiple unfolded proteins [65]. The third pathway involves activating ATF6, which upon dissociation from Bip translocates to the Golgi apparatus and undergoes proteolytic cleavage to become active. Activated ATF6 then enhances the expression of chaperone proteins like Grp78 (Bip) and C/EBP homologous protein (CHOP/GADD153) to manage unfolded proteins by either degradation or refolding [66,67].

**Figure 7 biology-12-01119-f007:**
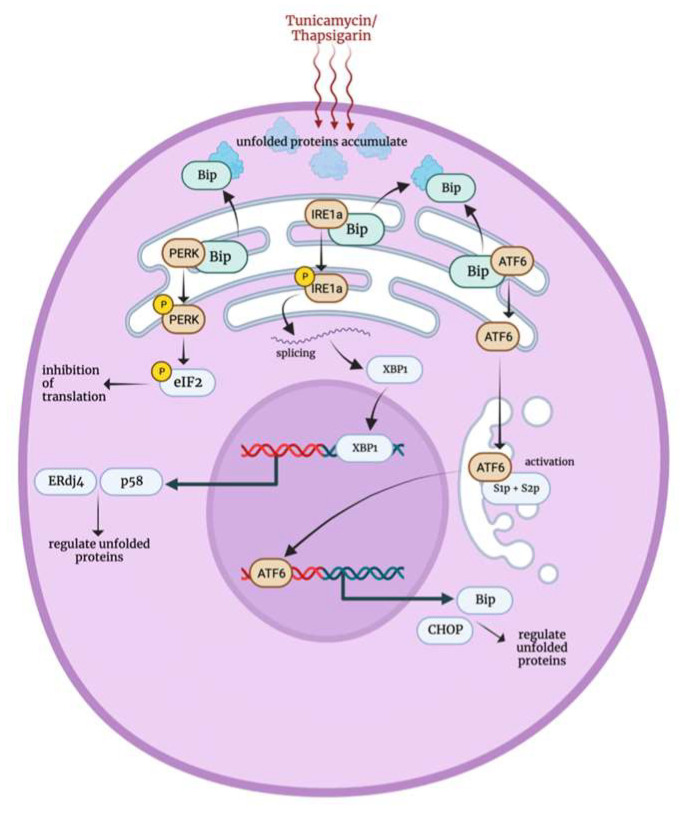
Endoplasmic reticulum stress response mechanism [59,60,61,62]. Bip: binding immunoglobulin protein, PERK: PRKR-like endoplasmic reticulum kinase, IRE1a: inositol requiring enzyme 1 alpha, ATF6: activating transcription factor 6, eIF2: eukaryotic translation initiation factor 2, XBP1: X-box binding protein 1, S1p and S2p: site 1 and 2 proteases, CHOP: C/EBP homologous protein.

## 4. Stress Factors Role in PSCs Self Renewal and Survival

Maintaining the viability and pluripotency of stem cells poses a significant challenge in stem cell research. Numerous investigations have focused on developing protocols to enhance the survival rate and sustain their pluripotent state. In this discourse, we discuss the impact of various stressors including mechanical, chemical, oxidative, and heat shock stress on the survival and pluripotency of PSCs.

In a pioneering investigation, the influence of gravity on the viability and self-renewal capability of iPSCs was examined. Murine iPSCs were loaded into a bioreactor installed on a spacecraft that embarked on a 14-day mission to outer space, subjecting the cells to genuine microgravity (μg). Concurrently, control cells were maintained in an identical device on Earth, experiencing the gravitational force of 1 g [68]. iPSCs derived from Oct4-GFP reporter mice exhibited enhanced regenerative capabilities when exposed to μg compared to cells cultured at 1 g. While both μg and 1 g iPSCs presented gradual growth, those cultured under μg conditions displayed a wider spread and extensive outward migration. The outspread iPSCs formed new clusters, accompanied by an increase in Oct4-GFP expression, which was proportional to colony size expansion. On the other hand, in the 1 g culture, only few cells migrated beyond the original colony, and as cell density increased, there was a notable decrease in Oct4-GFP expression. This decline reached a point where the recovery of Oct4 expression after overgrowth became unachievable. Conversely, iPSCs exposed to microgravity conditions maintained a robust Oct4 expression. However, after 6 days of microgravity culture, the Oct4-GFP fluorescent signal abruptly ceased and did not recover after continued growth [68].

In a comprehensive investigation focused on enhancing the survival and maintenance of hPSCs, various pharmacological agents were assessed for their effectiveness. Among the agent combinations tested, one particular mixture named CEPT, comprising Chroman 1, Emricasan, Polyamines, and Trans-ISRIB, stood out as the most potent. Notably, the presence of drugs from two distinct classes, namely ROCK inhibitors (e.g., Chroman 1, Y-27632, Blebbistatin) and pan-caspase inhibitors (e.g., Emricasan and Q-VD-OPh), proved beneficial in promoting cell survival [69]. ROCK, which stands for Rho-associated coiled coil kinase, serves as a critical regulator of myosin light chain phosphorylation and other cytoskeleton proteins. The activation of myosin, in turn, triggers interactions with actin, leading to cellular shrinkage, membrane blebbing, and cellular fragmentation during apoptosis [70]. Prior research had already indicated that silencing both ROCK1 and ROCK2 in human ESCs could enhance cell survival. Cells with knocked-down ROCK exhibited increased cloning capacity and strong Oct4 expression compared to control cells [71]. In the mentioned study, when hPSCs were exposed to the CEPT mixture for 24 h during routine cell passaging experiments, the vast majority of cells demonstrated remarkable traits: they remained undifferentiated, expressed essential pluripotency markers (Oct4, Sox2, Nanog, and alkaline phosphatase), and maintained normal karyotypes even after multiple passages [69].

To assess the protective effects of CEPT against chemical stressors, hPSCs were treated with Etoposide, which is a chemotherapeutic agent known to cause DNA damage. As the concentration of etoposide increased, the level of phosphorylated γH2AX, a marker of DNA damage (found in high levels during replication stress) [72], also increased. However, CEPT treatment reduced the effects of etoposide on hPSCs and prevented the presence of phosphorylated γH2AX, confirming CEPT’s protective role against chemical stressors [69]. Additionally, CEPT improved cell survival during the electroporation of dissociated hPSCs [69].

Further investigation was during single-cell dissociation, where cells were treated with DMSO, Y-27632 or CEPT. DMSO-treated cells exhibited signs of cellular stress, such as blebbing, while Y-27632-treated cells showed reduced stress but still displayed actin stress fibers. In contrast, CEPT-treated cells exhibited normal morphology with no signs of stress. Furthermore, CEPT-treated iPSCs showed an increased expression of tight junction protein 1 (TJP1 or ZO-1) and other related proteins like epithelial cadherin (CDH1), paxilin (PXN), and annexin A1 (ANXA1), which have a role in preventing cellular apoptosis. These findings indicate a lower level of stress and faster recovery compared to DMSO- or Y-27632-treated iPSCs [69].

In addition, the effect of CEPT was evaluated in cloning experiments, where CEPT-treated hPSCs generated a significantly higher number of clones compared to Y-27632-treated cells. Similar results were observed when comparing CEPT-treated cells to CloneR-treated cells; CloneR is a reagent specifically designed for single-cell cloning [69].

Moreover, CEPT treatment facilitated the formation of embryonic bodies (EBs) with the highest number and least cellular debris compared to other treatments (DMSO, Y-27632, and Chroman 1). These CEPT-treated EBs showed an increased expression of PAX6, Brachyury, and SOX17, indicating their capacity for multi-lineage differentiation [69].

Additionally, CEPT showed promising results in the cryopreservation and long-term storage of hPSCs. CEPT-treated cells, including hESCs, iPSCs, and astrocytes, exhibited significantly improved viability after 12 months of cryopreservation and thawing compared to cells treated with DMSO, Y-27632, RevitaCell (supplement that supports single-cell cloning), CloneR, or SMC4 (small molecule cocktail of 4 inhibitors) [69].

Furthermore, one study investigated the impact of osmotic stress on the viability of iPSCs and its influence on stress granule (SG) formation. SGs are membraneless cytoplasmic organelles that belong to a class of granules known as ribonucleoprotein (RNP) complexes. These granules are formed in response to various stress-inducing conditions, including oxidative stress [73]. During such conditions, the translation of mRNAs that produces repair-associated proteins is prompted, while the translation of mRNAs encoding other proteins is inhibited at the time of stress, and mRNA transcripts are sequestered within these SGs. This sequestration mechanism safeguards the mRNA molecules from degradation until the stressful conditions are resolved [74]. Therefore, SGs play a critical role in maintaining cell survival.

In a comparative study, commercial hiPSCs/IMR90-1 and non-pluripotent SH-SY5Y cells were subjected to different concentrations of sodium chloride (NaCl) treatment ranging from 0 to 400 mM for a duration of 1 h. Both cell lines exhibited SG formation, which increased with higher NaCl concentrations. This increase was evident through the progressive elevation of SG markers, G3BP, and LIN28. However, an interesting finding was that hiPSCs/IMR90-1 did not show any SG formation at 400 mM NaCl, unlike SH-SY5Y cells. Throughout the different NaCl treatment concentrations, hiPSCs/IMR90-1 maintained comparable expression levels of pluripotency markers, including Nanog, Oct4, and Sox2. Furthermore, hiPSCs/IMR90-1 exhibited preserved cell morphology, while SH-SY5Y cells showed signs of cell rounding and membrane shrinking. Similar observations were made when cells were treated with sorbitol instead of NaCl. Notably, there was no significant difference in the number of apoptotic cells between the NaCl-treated hiPSCs/IMR90-1 groups, as determined by propidium iodide (PI) and annexin V staining that are used for apoptosis detection. In contrast, SH-SY5Y cells exhibited increased features of late apoptosis as the NaCl concentration increased. Additionally, caspase 3 activation (an indicator of active apoptosis) was not detected in hiPSCs/IMR90-1 at 400 mM NaCl, suggesting that the absence of SG formation in these cells is not attributed to apoptosis activation [75].

To identify the pathways activated under hyperosmotic stress, Ingenuity Pathway Analysis (IPA) was conducted. In hiPSCs/IMR90-1, the highest-scoring IPA protein network was associated with cellular organization and assembly, cell-to-cell signaling, and reproductive system development and function. Conversely, the top-scoring IPA protein network in SH-SY5Y cells was associated with RNA damage, repair, post-transcriptional modifications, and protein synthesis. These findings highlight the distinct survival mechanisms activated in each cell group when subjected to hyperosmotic stress [75].

Furthermore, the impact of oxidative and heat shock stresses on SG formation in iPSCs was investigated. hiPSCs (IMR90-1) were exposed to different concentrations of sodium arsenite (SA) or hydrogen peroxide (H_2_O_2_) to induce oxidative stress, or they were subjected to various temperatures to induce heat shock stress. The results demonstrated that only SA and heat shock treatments induced SG formation, while H_2_O_2_-treated and control cells did not exhibit SG formation. Specifically, the formation of SGs in response to SA was concentration-dependent, with an increase in the number of SG-positive cells as the SA concentration increased. In contrast, iPSCs subjected to temperatures of 37 °C or 40 °C did not display any SG formation, whereas an elevation to 42 °C induced SG formation. These cells did not exhibit signs of apoptosis or differentiation. However, further temperature increase to 45 °C led to a decrease in SG formation and immediate apoptosis. Interestingly, H_2_O_2_ treatment did not induce SG formation in iPSCs, even at a high concentration of 2 mM. The cellular morphology of iPSCs following treatment (125 μM SA or 42 °C exposure) was compared to untreated cells, revealing no discernible differences in cellular morphology among the three groups. Therefore, the exposure to oxidative and heat shock stress at these levels did not alter the cellular morphology of iPSCs [76]. In addition, the expression of pluripotency markers (Oct4, Sox2, Nanog, Klf4, L1td1, and Lin28A) was evaluated under these stressful conditions. In cells treated with SA, a decrease in expression was observed specifically for Nanog and L1td1. Similarly, H_2_O_2_-treated cells exhibited a decrease in Sox2 expression and underwent apoptosis. In contrast, cells exposed to a temperature of 42 °C demonstrated a downregulation of all pluripotent markers [76].

## 5. Stress Factors Role in iPSCs Reprogramming and Differentiation

Since the discovery of iPSCs, somatic cell reprogramming has garnered significant interest in the field of stem cell research, as it aims to achieve functional pluripotent cells. However, this process remains inefficient due to various factors that impact its reproducibility and the quality of the generated iPSCs. Several factors have been identified as contributors to this inefficiency, including the delivery system of transcription factors, reprogramming target genes, donor cell type, and culture conditions [1]. Recently, the role of extrinsic stressors, including hypoxia, chemical stress, and mechanical stress, has gained attention in stem cell reprogramming due to their influence on the regulation of transcription factors and gene expression via the stress response mechanisms (Table 2) [7].

The utilization of high oxygen conditions has demonstrated its capacity to enhance differentiation processes. Notably, hiPSCs have been successfully differentiated into insulin-producing cells or endocrine progenitors by subjecting them to high oxygen conditions. This achievement was validated through immunofluorescence analysis, which confirmed the presence of C-peptide and glucagon positive cells, in addition to a noteworthy increase in INS gene expression that was observed in these cells [77]. In another line of iPSCs, differentiation into NGN3-positive cells was achieved within a high oxygen environment. This was accompanied by a substantial upregulation of NGN3 gene expression, indicating the successful generation of endocrine progenitor cells. Concurrently, the expression of VEGFA and HES1 was decreased, indicating a reduction in HIF-1α activity. Further investigation revealed that the Wnt signaling pathway, which is implicated in the regulation of cell fate determination, was activated in iPSCs subjected to high oxygen levels. Interestingly, the treatment of iPSCs exposed to high oxygen conditions with dickkopf-1 (DKK-1), a well-known Wnt antagonist, resulted in a decrease in NGN3 expression [77].

On the other hand, the exposure of hiPSC-derived liver buds (hiPSC-LB) to hypoxia has been shown to facilitate the differentiation process toward hepatocytes and cholangiocytes. The culture of hiPSC-LB under different hypoxic conditions, including excess, mild, and ambient oxygen levels, revealed distinct effects. Notably, mild hypoxia (10% oxygen) emerged as the optimal condition for maintaining the stromal lineage within the liver buds. Moreover, cells cultured under mild hypoxia exhibited the highest level of albumin secretion compared to those subjected to excess or ambient hypoxia. Additionally, vitronectin levels were found to be higher in cells treated with mild hypoxia compared to those treated with ambient hypoxia. Furthermore, liver buds cultivated under mild hypoxia conditions demonstrated a notable increase in CYP3A4 activity and urea production [78].

Further studies have shown that hypoxia plays a facilitating role in the proliferation of iPSCs undergoing differentiation into cardiomyocytes. Notably, iPSCs cultured under 4% oxygen levels demonstrated a significantly higher total cell count and expansion fold compared to cells cultured under atmospheric oxygen levels. In addition, cells cultured in hypoxic conditions displayed a positive expression of enhanced green fluorescent protein (eGFP that is translated under the control of cardiac-specific α-myosin heavy chain (αMHC) promoter), and they began beating by day 7 of culture. In contrast, cells exposed to normal oxygen levels exhibited only a limited number of eGFP-positive cells, and the occurrence of beating was rare throughout the culture period [79].

In contrast, a study revealed that under hypoxic conditions, the rate of neuronal differentiation of hiPSCs decreased. The hiPSCs were cultured in three different oxygen and carbon dioxide conditions: 20% O_2_ with 3% CO_2_, 20% O_2_ with 5% CO_2_, and 5% O_2_ with 5% CO_2_. The expression of neuronal markers, Pax6 and Sox1, exhibited the highest upregulation in the 20% O_2_ with 3% CO_2_ condition, while it was least pronounced in the 5% O_2_ with 5% CO_2_ condition. Similarly, the expression of motor neuron markers, oligodendrocyte transcription factor 2 (OLIG2) and neurogenin 2 (NGN2), was significantly increased in both cell groups cultured at 20% O_2_ compared to those cultured at 5% O_2_ [80]. These findings suggest that hypoxia may have a role in downregulating the neuronal differentiation of hiPSCs, indicating the importance of oxygen levels in the culture environment for the successful differentiation of hiPSCs into neuronal lineages.

Recent studies have explored the impact of gravity as a stress factor on hiPSCs. When hiPSCs were cultured under simulated microgravity conditions, they exhibited a loss of typical stem cell morphology, which was accompanied by a substantial increase in mesoderm posterior BHLH transcription factor 1 (MESP1) gene expression. Moreover, a remarkable 61% of the cells in culture displayed positive markers indicative of cardiac mesoderm induction, namely KDR and PDGFRα. In comparison to cells cultured under standard gravity, the microgravity-exposed cells maintained elevated levels of CD13, which is a marker for cardiac mesoderm [81].

Similarly, the impact of gravity on the differentiation process was investigated through the utilization of mouse EBs that were generated from either OCT4 reporter miPSCs or αMHC reporter miPSCs. The generated EBs from both types of iPSCs have showed a significant decrease in pluripotency markers and increase in differentiation markers in cultures maintained under microgravity conditions in comparison to those maintained under ground gravity. These findings highlight that iPSCs can undergo complete differentiation into myocardial progenitors more efficiently under microgravity conditions, suggesting an accelerated differentiation process in the absence of gravity [82].

## 6. Conclusions

This review highlights the significant impact of stress factors on PSC fate, with a particular focus on oxygen levels, which have been extensively studied. Intriguingly, different oxygen concentrations yielded distinct outcomes in PSC differentiation. For instance, hiPSC-derived liver buds (hiPSCs-LB) differentiation into hepatocytes was more successful at 10% oxygen compared to 2% oxygen [78], while iPSCs demonstrated optimal cardiomyocyte differentiation at 2% oxygen [79]. On the other hand, exposing mTSCs to 2% oxygen increased the expression of multipotency markers [9], whereas mTSCs cultured at 0.5% oxygen showed enhanced differentiation markers [10]. Both mTSCs and hTSCs exhibited increased proliferation at 2% oxygen, as indicated by elevated cyclin B expression compared to cells cultured at 20% oxygen [9]. Conversely, culture at 0.5% oxygen led to a decreased growth rate [10]. Moreover, hTSCs cultured at 20% oxygen experienced cell cycle arrest, as evidenced by an increased expression of p21Cip/Waf1 [10]. Notably, hiPSCs cultured at 60% oxygen displayed reduced HIF-1α activity and differentiated into pancreatic endocrine cells [77]. These findings suggest that mild hypoxia within the range of 2–10% oxygen may play a crucial role in both differentiation and increased proliferation. Atmospheric oxygen levels (20%) appear to hinder cell proliferation, yet intriguingly, excess oxygen (60%) promotes specific types of differentiation. Consequently, further research into the optimum oxygen levels for different cell line differentiations can deepen our understanding of these observations.

Investigations of PSC responses to other stress factors, such as osmotic, oxidative, and heat shock stress, revealed more lethal outcomes. ESCs exposed to sorbitol, for instance, showed increased pSAPK expression and ultimately underwent apoptosis in a dose-dependent manner [11,16]. However, in hiPSCs exposed to sodium chloride or sorbitol, stress granule formation was observed, preserving cell morphology and pluripotency marker expression, and no apoptosis occurred [75]. Additionally, the exposure of ESCs to oxidative stress that occurred due to zearalenone treatment resulted in an increased expression of apoptosis markers [13], whereas hiPSCs exposed to sodium arsenite exhibited stress granule formation with decreased pluripotency marker expression [76]. Interestingly, H_2_O_2_, another oxidative agent, did not induce stress granule formation in hiPSCs and underwent apoptosis [76]. Heat shock stress at 45 °C downregulated pluripotency markers in hiPSCs [76]. Thus, conducting additional studies to investigate the various cellular stress responses and compare their influence on PSC fate can help us gain a deeper understanding of why PSCs undergo specific changes under specific stressors, such as stress granule formation.

Recent studies have demonstrated positive outcomes when exposing iPSCs to mechanical stress. Real microgravity, for example, increased iPSCs’ regenerative capacity [68] and played a role in iPSCs and mouse EBs differentiating into cardiomyocytes [81,82]. Conversely, exposing ESCs to shear stress resulted in decreased proliferation and increased expression of MAPK8/9, which plays a role in inducing apoptosis [12]. Therefore, in-depth investigations comparing the effects of different types of specific mechanical stressors can help us better comprehend the diverse cellular stress responses activated in PSCs.

To summarize, our current understanding of the effects of stressors on stem cells remains incomplete, but several common associations have emerged from published experiments. Mild hypoxia appears to promote differentiation and cell proliferation in PSCs. Interestingly, hyperoxia also seems to facilitate the differentiation PSCs. Other stressors, including osmotic, shear, oxidative, and heat shock stress, have been shown to decrease cell proliferation and, at certain levels, induce apoptosis. Furthermore, the absence of gravity has been observed to accelerate the differentiation of PSCs and concurrently increase the regeneration capacity and colony size of PSCs. Given these findings, further exploration of the role of stress factors in stem cell fate determination and proliferation is necessary to fully understand the underlying mechanisms and harness their potential for stem cell maintenance and differentiation.

## Figures and Tables

**Figure 1 biology-12-01119-f001:**
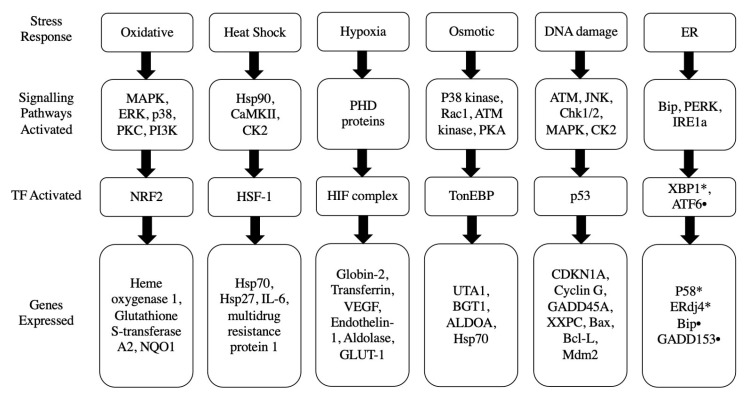
Stress response mechanisms and their associated protein sensors, transcription factors and target genes. TF: transcription factors, ER: endoplasmic reticulum.

**Table 1 biology-12-01119-t001:** Stress factors and their effect on embryonic stem cells growth.

Stress		Cell Type	Mechanism	Title 4	Ref.
Hypoxia (O_2_)	2% 20%	Mouse TSC ^1^	pSAPK, CDX2, Id2, Errβ increased	Multipotency	[9]
	0.5%		pSAPK, Gem1, Tpbpa, Hand1 increased	Differentiation	
	2%	Human TSC	Cyclin B + cell count increased	Proliferation	[10]
	20%		p21Cip1/Waf1 increased	Cell cycle arrest	
Osmotic	50 mM sorbitol	Human TSC + mouse TSC	pSAPK increased + decreased cell count	Decreased proliferation	[11]
	150 + 200 mM sorbitol		pSAPK increased + No cell growth	Stress marker	
	400 + 600 mM sorbitol		pSAPK increased	Apoptosis	
Mechanical	1.2 dynes/cm_2_ of shear stress	Mouse embryos	MAPK8/9	Apoptosis	[12]
Oxidative	Zearalenone 1, 2 and 4 μg/mL	hESC-derived EBs ^2^	p53, Bax, caspase-9 and caspase-3 + Bax/Bcl-2 ratio increased	Apoptosis	[13]
	Paraquat 75–100 μM	NT2 cells ^3^	Pax6, Gfra1, Hoxa1 and Ncam increased	Differentiation	[14]

^1^ TSC: trophoblastic stem cells, ^2^ hESC-derived EBs: human embryonic stem cell-derived embryonic bodies, ^3^ NT2 cells: human embryonal carcinoma Ntera2 cells.

**Table 2 biology-12-01119-t002:** Stress factors and their effect on iPSCs differentiation.

Stress		Cell Type	Markers Expressed	Directed Cell Type	Ref.
Hyperoxia	60% O_2_	hiPSCs ^1^	INS increased	Islet endocrine cells: C-peptide and glucagon positive cells	[77]
			NGN3 and VEGFA increased HES1 decreased	Ductal epithelium	
Hypoxia	10% O_2_	hiPSC-LBs ^2^	Albumin, vitronectin and urea increased CYP3A4 activity increased	Hepatocytes and Cholangiocytes	[78]
	4% O_2_	Murine iPSCs	eGFP and cell number increased beating activity detected	Cardiomyocytes	[79]
	20% O_2_ + 3% CO_2_	hiPSCs	Pax6, Sox1, OLIG2 and NGN2 increased	Motor neuron progenitor cells	[80]
	20% + 5% CO_2_				
	5% O_2_ + 5% CO_2_				
Gravity	Microgravity (μg)	hiPSCs	MESP1 increased KDR, PDGFRa, CD13 increased	Cardiomyocytes	[81]
		Mouse iPSC-EB ^3^	GFP increased	Myocardial progenitor cells	[82]

^1^ hiPSCs: human-induced pluripotent stem cells, ^2^ hiPSCs-LBs: human-induced pluripotent stem cell-derived liver buds, ^3^ mouse iPSC-EB: mouse-induced pluripotent stem cell-derived embryonic bodies.

## Data Availability

The data supporting the findings of this review are openly available at https://pubmed.ncbi.nlm.nih.gov/, accessed on 3 December 2022.

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
