# Peer review of "Stress Factors as Possible Regulators of Pluripotent Stem Cell Survival and Differentiation"

_biology, 2023, doi:10.3390/biology12081119_

Round 1

Reviewer 1 Report

The manuscript titled "Stress Factors as Possible Regulators of Pluripotent Stem Cell Survival and Differentiation" provides a comprehensive review of the stress response mechanisms in induced pluripotent stem cells (iPSCs). The authors have done a commendable job of compiling and summarizing the existing literature on this topic. The manuscript covers various aspects such as the induction of pluripotency, different stress response pathways, and the effects of specific stressors on iPSCs. Overall, the manuscript is well-written and informative. Please find below my specific suggestions for revision:

·        Include more recent references:

            The majority of references cited in the manuscript are from before 2019. Given the rapidly evolving field of iPSC research, it is important to include more recent references to ensure that the review reflects the most up-to-date knowledge. Incorporating recent studies would strengthen the scientific validity and relevance of the manuscript.

· Provide more critical analysis and discussion:

        While the manuscript effectively summarizes the existing literature, it would greatly benefit from a more critical analysis and discussion of the findings. This could involve comparing and contrasting different studies, identifying gaps in the current understanding, and proposing future research directions. Adding more critical insights would enhance the depth and originality of the review.

Overall, the manuscript has the potential to be a valuable resource in the field of iPSC research.

Reviewer 2 Report

The manuscript by Darwish et al., represents a timely important topic highlighting the stress factors and pathways that affect pluripotent stem cells in vitro. The manuscript is well written. However, there are few comments suggested that could improve the paper. 

1- In line 76, the authors could change the section title to be as the following " The Role of stress factors in Embryonic Growth and Proliferation" This would improve the readability. 

2- The title of Table 1 needs to be amended as the stem cells highlighted in the table is not focused only on embryonic stem cells but also include multipotent stem cells and placental mesenchymal stem cells. Please also consider amending the sentence in line 90-92 accordingly. 

3- In section 4, the authors reviewed thoroughly the role of stress factors in the self-renewal and viability of PSCs. It would be good to link the signaling pathways with the stress factors mentioned in this section. Also, Rock inhibitor was reviewed in this section and it is well known small signaling molecule used to improve the survival of PSCs and decrease stress. Thus, more info about rock inhibitor and its role and signaling pathways could be added in this section. By the end of this section, the authors mentioned that previous studies reported a decrease in the expression of pluripotency genes in response to stress factors, does this mean the cells are undergoing apoptosis or they are going to differentiate? 

4- In section 5, the authors could also review the effect of stress factors such as oxygen levels on neural differentiation.

Reviewer 3 Report

This review is well written and easy to understand. I suggest this review be accepted for publication.

Round 2

Reviewer 1 Report

The improvements reflected in the revised version of the manuscript determine my conclusion of acceptance in the present form.